# Development and Validation of a Low-Cost Device for Real-Time Detection of Fatigue Damage of Structures Subjected to Vibrations

**DOI:** 10.3390/s23115143

**Published:** 2023-05-28

**Authors:** Agnese Staffa, Massimiliano Palmieri, Giulia Morettini, Guido Zucca, Francesco Crocetti, Filippo Cianetti

**Affiliations:** 1Department of Engineering, University of Perugia, Via G. Duranti 93, 06125 Perugia, Italy; 2Aeronautical and Space Test Division, Italian Air Force, Via Pratica di Mare, 00040 Pomezia, Italy

**Keywords:** vibration fatigue, early fault detection, real-time fatigue damage

## Abstract

This paper presents the development and validation of a low-cost device for real-time detection of fatigue damage of structures subjected to vibrations. The device consists of an hardware and signal processing algorithm to detect and monitor variations in the structural response due to damage accumulation. The effectiveness of the device is demonstrated through experimental validation on a simple Y-shaped specimen subjected to fatigue loading. The results show that the device can accurately detect structural damage and provide real-time feedback on the health status of the structure. The low-cost and easy-to-implement nature of the device makes it promising for use in structural health monitoring applications in various industrial sectors.

## 1. Introduction

Monitoring the structural behavior of mechanical components and systems has been widely used in engineering for many years. Since damage monitoring techniques offer the significant advantage of reducing the risk of unexpected failures, leading to increased safety and decreased maintenance and repair costs [1,2]. As a result, it is applied in several fields of engineering, for example, damage monitoring technique and/or early fault detection methods are widely used on civil structures, such as bridges, dams or any infrastructures that can attempt the safety of people [3,4]. In mechanical engineering, one of the most explored sectors for health monitoring is wind turbines [5,6,7]; however, it is easy to find applications in several different sectors, such as automotive or aerospace [8,9].

Due to the increasing use of engineering systems in harsh environmental conditions for all sectors of engineering, active control techniques have emerged to minimize fatigue damage and reduce structural loads [10]. For this reason, it is easy to find in literature several techniques for the health monitoring of structures. The most commonly used approach for dynamic structures is to monitor any change in the dynamic response of the system. This kind of analysis is generally made through several techniques, such as ultrasonic, acoustic emission, or vibration analysis. All these techniques involve analyzing data acquired by sensors positioned on the structures and observing any changes in the natural frequencies, damping, or modal shapes [11,12,13,14]. In recent years, the diffusion of computer vision, artificial intelligence, and machine learning has further extended the opportunities offered by damage monitoring methods to the damage localization and quantification [15,16,17]. For example, Yan et al. [18] used the classical vibrational theory in combination with the artificial intelligence to increase the accuracy of vibration-based structural health monitoring techniques. In this framework, Alavi et al. [19] proposed an innovative approach combining the finite element method (FEM) and probabilistic neural network (PNN) based on Bayesian decision theory for damage detection. Ngoc et al. [20] proposed a novel approach for damage detection in structures based on a combination between artificial neural network and cuckoo search algorithm demonstrating their efficiency, especially in terms of computational effort, to detect structural damage using a calibrated numerical model of a steel beam and a large-scale truss bridge. Kathir et al. [21] proposed a technique based on artificial neural network (ANN) in combination with particle swarm optimization (PSO) aimed at damage quantification in laminated composite plates using Cornwell indicator (CI). Feng et al. [22] have developed a new approach, based on approximate Bayesian computation with subset simulation (ABC-SubSim), for the approximation of a model using modal data.

An innovative approach for the detection of fatigue damage has been proposed by Patil et al. [23]. Their approach merges the vibration-based method with the non-destructive ultrasonic C-scan method. The aim is to effectively detect, localize, and measure the extent of damage in carbon composite fiber-reinforced structures subjected to impacts. An interesting activity is the one of Tang et al. [24] who proposed a method that exploits the photomicroscope to detect the small crack propagation in a very high-cycle fatigue regime with reasonable time-cost. Bhowmik et al. [25] proposed a novel reference-free approach for identifying structural modal parameters using recursive canonical correlation analysis. The method utilizes a first-order eigen-perturbation approach to determine the normal modes of a vibrating system at each instant of time, and then used these modes to detect a potential damage throughout the modal assurance criterion. Recently, Cianetti et al. [26,27] proposed a damage estimation method able to monitor the accumulated fatigue damage in a structure or potential damage in real-time. This method involves applying a cycle counting method and the Palmgren–Miner damage accumulation rule [28] to a moving window of the signal. The significant advantage of this technique is that it can determine the cumulative damage on the entire structure with a single measurement, based on a numerically-determined relationship. Indeed, assuming that the component is linear, it is possible to determine the system’s frequency response functions between the forcing function and any physical quantity used to evaluate actual and/or potential damage (potential if calculated on a different quantity than stress), for any point in the structure, using a finite element numerical model. However, the potential of the approach proposed by Cianetti et al. [29] has only been evaluated numerically, and the possibility of implementing this methodology in a real-time physical device capable of performing the required tasks has not yet been experimentally tested.

For this reason, in this work, the monitoring technique proposed by Cianetti et al. [29] has been implemented in an ad-hoc device, using easily available and low-cost components. A high-performance but low-cost processor and analog-to-digital converter were used for the hardware, while the damage calculation algorithm was developed in Python. Once the device was fabricated, it was experimentally tested on a Y-shaped component [30] subjected to vibration tests using different excitation profiles. During the various tests, the potential damage was evaluated in real-time up to the predicted failure time. The validation of the device and algorithm was then performed by comparing the estimated real-time lifetimes provided by the system with those obtained by monitoring the drop in natural frequency in post-processing analysis [31]. The results obtained demonstrate that the device provides sufficiently accurate results, with a percentage error committed always below 20%.

Therefore, the result of this work enables the problem of identifying possible fatigue failure in a mechanical component to be addressed using a very simple algorithm and low-cost electronic components. Although the device was tested on a simple laboratory component, its potential is not diminished, as it can be applied to any mechanical component or system.

This paper is organized as follows: Section 1 describes the traditional approach used to calculate fatigue damage in the time domain at a theoretical level. Section 2 presents the monitoring technique that was then implemented in the fabricated device. Section 3 describes the optimization process followed to define the acquisition parameters. Section 4 describes the device from both a hardware and software perspective, while Section 5 illustrates the experimental validation of the device and the results obtained. Section 6 presents the conclusions.

## 2. Theoretical Background

The calculation of mechanical component damage is performed to verify the material’s resistance when subjected to cyclic variations in time, which, although lower than the maximum load, can still lead to material failure. The quantitative characterization of the phenomenon is carried out by dividing the load history into cycles described by mean and alternating values. Each cycle, based on its amplitude and number of occurrences, contributes proportionally to the component’s damage. Although several methods exist in literature for assessing fatigue damage [28,32], this work utilizes Palmgren–Miner’s accumulation law [33]. This law states that each cycle contributes a damage factor to the overall life of the component, equal to the percentage of life consumed at those conditions, independent of the order of application of various cycles. If the cycle is characterized by a non-zero mean value, it lowers the material’s resistance curve. Correction is, therefore, necessary to account for the effect of the mean stress. One of the most used is Goodman’s correction [32], which returns a purely alternating stress $\sigma_{a,eqv}$ (Equation (1)):(1)$$\sigma_{a,eqv}=\frac{\sigma_{a}}{1+\frac{\sigma_{m}}{S_{ut}}}$$

At this point, it is possible to use the Wohler curve, which relates the amplitude of the cycle’s oscillation to the number of cycles at which fatigue failure will occur, to determine the number of cycles (Equation (2)) at which the material would withstand before failure:(2)$$N_{i}={\left(\frac{\sigma_{a,eqv}}{a}\right)}^{\frac{1}{b}}$$

In Equation (2), *a* and *b* represent the intercept and slope of the Wohler curve, respectively. Once this value is determined, it is compared with the number of cycles, *n_i_* at that level, recorded in the loading history. This comparison allows for the calculation of the damage portion caused (Equation (3)) by the *i*-th cycle:(3)$$D_{i}=\frac{n_{i}}{N_{i}}$$
when *D_i_* reaches unity, fatigue failure of the component occurs.

The load history of the component under examination, as in most cases, is subjected to random fluctuations and high-frequency oscillations, where it is not possible to uniquely identify each cycle with its mean and alternating value. Therefore, it is necessary to use cycle counting methods [34]. In this work, a biparametric method that takes into account both the amplitude and mean value, namely, the Rainflow counting method (RFC), is utilized [35].

The purpose of the method is to identify closed hysteresis load cycles that represent the material’s energy dissipated during oscillation, and thus is responsible for the component’s fatigue damage. For each identified cycle, mean, alternating values, and the number of cycles are determined and are typically represented in the Rainflow mean-range matrices and the load spectra.

The traditional method for calculating the damage of a mechanical component subjected to a typically oscillating load history consists of acquiring a signal instant-by-instant through sensors, in order to generate a load history that increases over time and evaluate the damage of that time history, from the initial instant to the current one, at each instant. Once the history is acquired, a cycle counting method, such as Rainflow counting can be applied to extract a load spectrum, in which the mean value and alternate value are reported for each extracted cycle. If cycles with non-zero mean value are obtained by the RFC algorithm, it is necessary first to adopt Equation (1) to calculate an equivalent uniaxial component. Finally, a damage accumulation rule, such as the Palmgren–Miner rule is applied to this load spectrum to obtain a value of damage up to the considered instant (Equation (4)):(4)$$D_{p}=\overset{m}{\underset{i=1}{\sum }}\frac{n_{i}}{{\left(\frac{\sigma_{a,eqv}}{a}\right)}^{\frac{1}{b}}}$$
where *m* is the total number of counted cycles.

This process is continuously applied until the excitation is removed. Figure 1 shows a flowchart of the traditional procedure for calculating the fatigue damage.

If the acquired temporal history is expressed in terms of stress, then the damage calculated with the method highlighted earlier represents the actual damage suffered by the component, and thus provides a precise indication of its residual life. However, the method presented can also be applied to a generic physical quantity. In this latter case, the computed damage does not have a physical sense (it can be used instead to highlight a situation when the structure is particularly excited). Therefore, if one has a fatigue resistance curve expressed in the unit of measurement of the analyzed quantity, such as the one reported in Equation (5), it is possible to obtain an indication of the residual life of the component also by measuring physical quantities different from stress:(5)$$x_{f}=aN^{b}$$
where *x_f_* represents the amplitude of the input signal, *a* and *b* represent the intercept and slope of the resistance curve of the parameter being monitored, expressed in the same unit of measurement as *x_f_*. If an appropriate fatigue resistance curve is not available, the calculation can still be performed, but what is obtained is not actual damage, but rather potential damage indicated as *D_p_* (Equation (6)):(6)$$D_{p}=\overset{m}{\underset{i=1}{\sum }}\frac{n_{i}}{{\left(\frac{x_{a_{i}}}{a}\right)}^{\frac{1}{b}}}$$

Therefore, the potential damage calculated is not indicative of the residual life of the component, but can be used as a parameter for comparing different load conditions, for example, to highlight whether a particular condition is potentially more damaging than another or to highlight any temporal instants in which the structure is particularly stressed. Whatever type of damage is calculated, i.e., real or potential, the traditional method involves applying the cycle counting method for the entire time, from the moment the component is installed to the moment the calculation is carried out, and thus presents the problem related to the large amount of data that increase over time. The elaboration of this large amount of data requires high-processing times, and this implies that evaluating damage in real-time with this type of approach is almost impractical.

## 3. Real-Time Estimation of Damage

The approach presented in the previous section is not effective with a real-time analysis as it would involve managing an enormous amount of data (especially for high-frequency signals). An innovative method for estimating damage, which is proposed by Cianetti et al. [29], is based on the same methodology as the classical method, the Rainflow counting algorithm, and Palmgren–Miner rule, but rather than performing the calculation at every instance on the entire acquired time history, it is performed on a moving window, and thus significantly reduces the computational effort. The proposed approach is shown in Figure 2.

In contrast to the classical method, the main advantage of the proposed method for the estimation of fatigue damage is that it involves analyzing only a small window of length ∆*T*. This means that the amount of data to be managed and processed is small and can be addressed with low-cost devices. Indeed, the Rainflow counting technique and Palmgren–Miner rule are applied only on the *i*-th window and the rest of the data are disregarded in order to not fill the memory. These data are needed to determine the potential instantaneous damage associated, as shown in Equation (7):(7)$$d_{p_{i}}=\overset{m_{i}}{\underset{k=1}{\sum }}\frac{{\left(n_{k}\right)}_{i}}{{\left(\frac{{\left(x_{a_{k}}\right)}_{i}}{a}\right)}^{\frac{1}{b}}}$$
where a and b are the parameters of the Wohler curve expressed in terms of analyzed physical quantities, $n_{k}\;$ is the number of the counted cycles from the *i*-th window, k is the generic cycle of the counted spectrum, $x_{ak}$ represents the alternate value of the acquired signal, and $m_{i}$ is the total number of cycles in the acquired signals.

Instantaneous damages *d_pi_* can be used to detect trends of increasing fatigue stress of the structure, which once critical thresholds are exceeded, signal a risky mode of operation, enabling prompt intervention activities. Once the potential fatigue damage associated with the *i*-th window is known, the accumulated total damage at a predetermined time instant *D_p_* can also be calculated through the Miner’s cumulative damage law as the sum of instantaneous damages *d_pi_* (Equation (8)):(8)$$D_{p}=\overset{i}{\underset{r=1}{\sum }}d_{p_{i}}$$

The “potential” damage provides an indication of the residual life of the machine, allowing for the monitoring of the achievement of the unitary value and avoiding fatigue failure.

### 3.1. Analysis of Criticalities

When deciding to work in real-time [29], critical issues often arise due to the need to provide instantaneous results, which may compromise the accuracy of the results. As indicated in the preceding paragraph, in order to calculate the damage of structures in real-time subjected to dynamic loading, it is necessary to work with windows rather than long time histories. Therefore, there are three issues to be analyzed and managed before implementing the real device and testing it.

In particular, three fundamental criticalities have been identified:Window duration;Residue management;Sampling frequency.

The window duration represents the acquisition and processing time of the portion of the signal that is actually analyzed in real-time. The duration of the window is determined exclusively by the lowest frequency content that needs to be observed. In fact, the window length should be equal to a multiple of the inverse of the lowest frequency (Equation (9)):(9)$$\Delta T=\frac{k}{f_{min}}$$

Using a window length that is shorter than indicated in Equation (9) carries the risk of missing crucial phenomena in the overall process, resulting in inaccurate outcomes. Therefore, the window length plays a vital role in calculating real-time damage. However, working with longer windows necessary to capture low-frequency phenomena presents a challenge when it comes to operating in real-time. Longer windows entail managing and processing a substantial amount of data, which is impractical in real-time application.

When performing cycle counting on short windows, there is a high risk of managing numerous residues that would not occur with very long time histories. However, as indicated earlier, to operate in real-time, it is necessary to use few data, and thus the residue management strategy becomes crucial in obtaining potential and/or cumulative damage values similar to those obtained by cycle counting on a long time history. To address the problem, four different methods have been considered [34,35,36,37,38,39]:Ignored residues;Residues counted as half a cycle;Residues carried over to the next window;Residues counted as full cycles in the hypothesis of repeating history.

In the first case, residues are completely ignored, and only the closed cycles are considered for the calculation of damage, resulting in an approximation that underestimates the final damage. The second approach consists of considering residues as half a cycle [34]. In the third case, the residues left open in the previous window are carried to the next one, with the aim of recreating the periodicity of the original entire history and allowing for the spontaneous closure of residues. The fourth method assumes that the loading history repeats itself identically to infinity, creating duplicates that give rise to repeated maxima and minima, which then correspond to complete cycles. This method generally tends to overestimate the damage as it considers each residue as a complete cycle rather than as a half cycle [39].

The sampling frequency of a signal determines the time interval between an acquisition and the next one, and thus depends on the maximum frequency to be observed. According to the Nyquist theorem, it should be at least two times the maximum frequency content of the signals [40]. Generally, to accurately sample high-frequency variations, the two times of the maximum frequency may not be sufficient and, for this, a sampling frequency ten times higher than the maximum frequency of the process is commonly used. However, even in this case, working with high sampling frequencies entails possible criticalities in operating in real-time, and thus analyses have been carried out to determine the optimal sampling frequency value.

The determination of optimal parameters for the minimum duration of the acquisition window, the residue management strategy, and the sampling frequency to work in real-time has been numerically addressed. The choice of optimal parameters and strategy was made by comparing the results obtained using various options with the proposed method and the results obtained with the conventional method. In particular, the potential damages obtained with the two methods (classical—proposed) and the mean-range matrices obtained with the Rainflow counting method were compared. Correspondence between these results indicates good accuracy of the results. To this end, various signal types were chosen and generated to cover a wide range of signals generally encountered by mechanical systems subjected to vibrations. Five different signal types have been identified for analysis:Non-stationary signals;Non-stationary signals in sections;Real signals with variable average;Wideband signals;Bimodal signals.

Non-stationary signals are random signals whose statistical parameters are not constant over time. This type of signal is very common in nature, and thus must necessarily be taken into account [41,42,43]. To test the algorithm, a real signal whose average value is different from zero in some sections was also chosen. In order to confer general validity to the algorithm, it is necessary to analyze signals with different shapes and distributions of power spectral density (PSD) to verify that the algorithm does not lose its effectiveness when working at certain frequencies, perhaps due to difficulties in the cycle counting phase. Therefore, a wideband signal with frequency content distributed over a wide range of frequencies and a bimodal signal, whose frequency content is distributed with a rectangular shape of varying amplitudes over a double frequency range, have been selected. As the purpose of the study is to evaluate the optimal parameters for signal analysis, the various comparisons were made by imposing the parameters of the fatigue resistance curve. These parameters were arbitrarily chosen and are α = 400 MPa and β = −0.3. The tests and comparisons were performed on all the considered signal types and are presented in detail only for one case. The results obtained for the remaining signals are synthetized at the end of each subsection since they are analogous to the detailed case.

#### 3.1.1. Variation of the Window Duration

In this first case, the results are reported for the non-stationary signal case, characterized by a non-zero power spectral density (PSD) in the frequency range from 28 to 36 Hz (Figure 3).

Initially, the damage was calculated using the classical procedure for the considered signal, taking into account the entire time history. The resulting damage, based on the chosen parameters, was found to be 0.1699. In the first testing phase, the signal was analyzed by varying the duration of the acquisition windows while fixing the other control parameters. Specifically, a sampling frequency of 720 Hz was imposed, which is 20 times higher than the maximum frequency of the signal. Furthermore, the residue management strategy recommended by the relevant technical standard was chosen, i.e., each residue was calculated as half a cycle. However, a different choice would lead to a very similar result since, when working with very long random temporal histories, the number of residues is negligible.

To validate the capability of the considered logic as the acquisition window duration varied, different durations were considered according to Equation (9), assuming various k-values. The results obtained from these series of analyses are reported in Table 1. Observing the last column, which reports the percentage error in damage with respect to what was obtained with the traditional method on the entire time history, it is noted that the error decreases below 2% only for window durations greater than 28 times that are computed considering only the minimum window.

This is even more evident in Figure 4, where the potential damages obtained for different window durations are compared to the traditional method.

The results show that a short window duration underestimates the final damage value (Figure 4), and there is an increase in residues, with a consequent increase in cycle dispersion. To ensure the obtained results, the Rainflow (mean-range) matrices were also compared, as shown in Figure 5.

For the other types of signals, similar results are obtained and synthetized in Table 2. For the analysis, a sampling frequency equal to 20 times the maximum frequency contained in the signal as well as an evaluation of the residuals as half a cycle and variation of the window duration were used.

As visible from Table 1 and Table 2, the optimally compromised window length that leads to cumulative damage values equal to those calculated on the entire history corresponds to a duration at least 30 times greater than the inverse of the minimum signal frequency. Indeed, with a window length equal to 30 times the inverse of the minimum frequency of the signal, the absolute percentage error between the fatigue damage computed with the proposed method and the one obtained considering the entire time signal is always around 1–2%.

#### 3.1.2. Variation of Residue Counting Strategy

The analysis of residual counting strategies was also carried out on all types of signals, but only the case of non-stationary signals (Figure 3) is reported in detail. Similar to what was carried out previously, the only variable parameter was the residual counting method, while keeping the other parameters, i.e., sampling frequency and Wohler curve, constant and fixing the window duration to the longest of the values in the previous paragraph, which corresponds to an error of approximately 0%. All four types of residual counting methods (described in Section 3.1) were tested. The results are summarized in Table 3.

A comparison between the cumulative damage calculated using the classical method and the method proposed here, with their respective strategies, is shown in Figure 6.

For the rest of the analyzed signal types, the results are shown in Table 4.

The analysis and the results in Table 3 and Table 4 showed that the best methods for obtaining cumulative damage values equal to those calculated from the entire loading history are those in which the residuals are treated as half a cycle or transferred to the next window. Adopting one of these two methods, the computed fatigue damage is similar to the one obtained considering the classical method. Indeed, the absolute percentage error between the estimated fatigue damage and the actual one has a mean value equal to 3.3% adopting the third residue management technique (move to next window) and a mean value equal to 1% using the latter (half cycle). As expected, the method in which the residuals are neglected underestimates the damage calculation, while the method in which they are counted as whole tend to overestimate it.

#### 3.1.3. Variation of Sampling Frequency

Once the optimal values of the two previous parameters were determined, an analysis was performed to determine the influence of sampling time variation. The case of a wideband signal was considered, characterized by a non-zero PSD in the frequency range from 50 to 200 Hz (Figure 7).

The cumulative damage value calculated using the classical method was found to be 0.9782. As for the previous cases, the other parameters were kept at the values which presented a minor damage calculation error in the previous analyses; therefore, the window duration was set to the greater value of *k*, described in Table 1 and Table 2, the residuals were treated as half a cycle, and the parameters of the Wohler curve were the same as those used in the previous paragraphs. To observe the effects of the variation of this parameter, according to the Nyquist theorem, which states that the sampling frequency must be at least two times the maximum frequency, the time intervals were defined as the inverse of multiples (k) of the maximum frequency contained in the signal, using the following formula (Equation (10)):(10)$$dt=\frac{1}{k\cdot f_{max}}$$

The results are summarized in Table 5.

The comparison of cumulative damage for the wideband signal between the various tests performed and the one obtained with the classical method is shown below (Figure 8).

For the purpose of completeness, the results obtained for the other considered signals are also reported in Table 6.

From the results shown in Table 5 and Table 6, it can be stated that, as already well-known from signal analysis, considering a frequency sampling equal to two times the maximum frequency to be observed is not sufficient to adequately represent all the variations of a high-frequency signal. Indeed, a frequency, at least, ten times higher than the maximum frequency of the signal is necessary. Assuming this value of frequency sampling, the error committed with the proposed algorithm is around 3–4%, while assuming a frequency sampling equal to two times the maximum frequency of the signal leads to a percentage error of over 50%.

In conclusion, from the carried-out analyses, the optimal values for calculating the damage in real-time appear to be: Duration of the analysis window equal to at least 30 times the minimum frequency, the residuals considered as half a cycle, and finally the sampling frequency equal to at least 10 times the maximum frequency. The experimental tests carried out to test the proposed algorithm are conducted using the optimal values listed earlier.

## 4. Hardware and Software Prototype for the Real-Time Monitoring of Fatigue Damage

Once the main acquisition parameters and the strategy to manage the residues were extracted by the Rainflow counting algorithm, thereafter the next step was to develop a physical prototype and an algorithm capable of computing the fatigue damage in real-time. In the following sections, a description of the hardware and software is provided.

### 4.1. The Hardware

The device used in this study comprises five essential elements, each serving a specific purpose. These include a sensor for signal acquisition, a signal conditioner for signal amplification, an AD converter to convert the analog signal into a digital one, a buffer board to store acquired data, and a processor for simultaneous data computation.

In this study, as a signal transducer, an accelerometer produced by PBC was used as the sensor. For this reason, a potential fatigue damage based on the acceleration response is computed according to what was stated in the previous section. Using an accelerometer, it was further necessary to use a signal conditioner. A conditioner model 480E09 produced by PBC was adopted. The Raspberry Pi AD/DA Expansion Board ADS1256, with eight channels at 24-bit and a sampling rate of 30 ksps, was used as the ADC. The WeAct Black Pill V2.0 board was chosen for the buffer board to store acquired data, while the Raspberry Pi 4 was used as the processor. These components were selected to minimize the total cost, in which excluding the sensor and signal conditioner is less than EUR 150. Figure 9 shows the final prototype of the device used in the study.

As shown in Figure 9, an additional shunt resistor between the ADC converter and the buffer board was required to insert a bias. This was needed since the selected ADC converter can read only positive values within the range of 0–3.3 volts. Therefore, the shunt resistor adds a bias, and the signal is only positive with a mean value different from zero but of known value equal to 1.75 volts. In this way, negative values are not lost. However, since as expected the acceleration response is generally a zero mean signal, the introduced mean value is removed when the signal is processed by the software.

### 4.2. The Software

Once the signal is accurately acquired by the device shown in the previous section, it was necessary to define an algorithm that is able to read the data, process them by applying a cycle counting technique and, at the same time, able to supply an indication of fatigue damage (real or potential) in real-time. To this aim, a Python algorithm has been implemented on the Raspberry Pi 4. This algorithm exploits the multi-threading logic. In this way, two threads can operate simultaneously. The first thread acquires real-time signal portions (windows), while the second calculates the fatigue damage of the previously acquired window, as explained in Section 2 (Figure 10).

As thread2 processes the data obtained by thread1, upon device start-up it immediately acquires a window outside the while-loop and directly conveys it to thread2. Meanwhile, thread1 acquires a second window, retaining it in memory through the buffer board and awaits thread2’s readiness to process new data. This ensures that all operations occur within the chosen processor, without the risk of data loss.

As shown in Figure 10, all operations occur within a while-loop which concludes when the algorithm’s computation of potential fatigue damage reaches a critical threshold. This threshold indicates that the component has accumulated damage deemed critical to the structure. The value is equivalent to the unit when the signal acquired is stress, but can differ when dealing with other physical quantities, such as accelerations, forces, etc. This value can be determined numerically by highlighting relationships between the measured signal and actual structural damage.

## 5. Experimental Validation of the Realized Device

In order to evaluate the ability of the device to operate in real-time for the monitoring of a fatigue failure induced by dynamic loads, vibration tests were performed on a simple laboratory component. The specimens were realized in additive manufacturing in PolyLactic acid (White-Pearl PLA produced by Ultimaker, Utrecht, The Netherlands). Figure 11 shows the used experimental setup. A detailed description of the printing parameters and the manufacturing process is provided in [31,44].

As shown in Figure 11, the Y-shaped specimen was excited with a series of random acceleration profiles through an electrodynamic shaker. The random profiles were defined by three different power spectral densities (PSDs) with varying frequency contents and RMS. The used PSDs have the following frequency ranges: 90–200 Hz, 80–250 Hz, and 100–400 Hz, while the used RMS values were 1 g, 3 g, and 2 g, respectively. Since the fourth natural frequency of the system is equal to 195 Hz, the excitation PSD is always defined around the natural frequency of the structure. Figure 12 shows the waveform of the adopted PSD.

According to the experimental setup shown in Figure 11, the parameter chosen to monitor the fatigue life of the Y-shaped specimen is the acceleration response measured on the arm of the sample. However, since the S-N curve of the sample was determined in a previous activity [30], it was of interest to use the same response acceleration and the known fatigue curve. To use this response acceleration and, at the same time, use the fatigue curve, it was necessary to determine a scale factor able to transform the acceleration measured on the arm of the specimen to the stress component at the failure point.

To this aim, an FE model of the sample has been realized, replicating the actual experiment condition (Figure 13a), and then it has been calibrated with the experimental data in terms of acceleration frequency response function (Figure 13b) [30]. Details of the FE model are provided in Appendix A.

Once a finite element model is calibrated, experimental data are viable and it is possible to use these data to obtain a frequency response function Hσy¨ between the base acceleration ($\ddot{y})$ and the stress component that causes the failure of the component. In fact, depending on the type of inputs used, only the fourth vibrating mode of the system is excited, which is the symmetrical bending of the arms (Figure 14a). This means that the point at which the test sample fails is easily identifiable and that the stress component that leads to the failure of the test piece is only the stress component due to the bending of the arms (Figure 14b) [30].

Therefore, once the two frequency response functions were addressed, the first between the acceleration on the arm ($\ddot{x}$) and the base acceleration ($\ddot{y}$) defined as Hx¨y¨, and the other between the stress component (σ) at the failure point and the base acceleration ($\ddot{y}$) defined as Hσy¨, it is possible to identify the frequency response function between the stress at the failure point and the acceleration measured on the arm as follows:(11)Hσx¨= Hσy¨· Hx¨y¨−1

The frequency response function Hσx¨ obtained is shown in Figure 15.

Although in this case the component is subjected to an uniaxial stress state induced solely by the bending of the arms, even in the case of a multiaxial stress state, what is proposed remains applicable. In fact, the developed device processes both an uniaxial alternating stress or an equivalent uniaxial stress time history obtained by a multiaxial synthesis criterion.

Assuming the validity of frequency response function, a scaling factor was determined in order to transform the acceleration measured on the specimen’s arm with the stress at the failure point. This scaling factor η was considered as the maximum of the frequency response function at the specimen’s resonance frequency (Figure 15). This factor is equal to 0.146 MPa/g. In this way, Palmgren–Miner’s rule can be directly applied for damage calculation, along with the known fatigue strength curve, as shown in Equation (12):(12)$$D=\overset{m}{\underset{i=1}{\sum }}\frac{n_{i}}{{\left(\frac{\eta \cdot {\ddot{x}}_{a,i}}{a}\right)}^{1/b}}$$
where ${\ddot{x}}_{a,i}$ is the response acceleration measured during the test.

Once the scale factor is known, it was able to perform the experimental tests using the method described previously in order to check the capabilities of the realized device. Therefore, vibration tests were performed using an electrodynamic shaker and the signal measured by the accelerometer on the specimen arm was simultaneously acquired and processed with the device and recorded with a traditional acquisition system. This last step was used for the evaluation of the actual fatigue life of the component in post-processing by monitoring the resonance frequency drop. The results obtained with the realized device are shown in Figure 16, where the cumulative damage curve is reported over time. Figure 16 also shows the threshold that indicates the component’s failure point. This threshold was defined as the unit according to Miner’s cumulative damage law.

To compare the results obtained from the realized device, the duration was evaluated in post-processing by monitoring the frequency drop. This method, widely used in vibration fatigue, involves monitoring the natural frequency of the component during the test. Indeed, the start of a fatigue fracture results in a variation of the stiffness of the system, and thus in a decrease in the natural frequency. Usually, a component is considered to be failed when the monitored natural frequency drops by 5% compared to its initial value [31]. The results obtained by monitoring the frequency drop for the conducted tests are shown in Figure 17.

The obtained results are then compared in Table 7 in order to better highlight the differences between what is obtained with the proposed device and the traditional frequency drop.

Comparing the estimated fatigue life obtained with the proposed device with the one obtained by monitoring the frequency drop, it is evident how the proposed device supplies good results, in agreement with those traditionally computed. The percentage error is always lower than 20%. As visible from Table 7, the device always anticipates the failure of the component. However, this is attributable to the fact that for the damage calculation, a scaling factor between the acceleration measured on the arm and the tensile stress equal to the value assumed by the magnitude of the frequency response function at resonance was used. However, considering that the maximum value of the frequency response function is conservative, the system is not excited only at the resonance frequency, but also with a wideband signal. This implies that the stress–acceleration relationship is not always equal to the maximum of the frequency response function, but can assume lower values, and thus the tensile stress would be lower than the one calculated.

However, the obtained results are in line with what can be obtained from traditional methods and represent a good starting point for the evaluation of fatigue damage in real-time using a low-cost device. In fact, the device allows for the evaluation of the damage of a whole structure with a single measurement point, exploiting a numerically calibrated model to define the correlation between the points where the outputs are measured and the critical points. In addition, this methodology would allow for the calculation of damage at points where it would not be possible to insert classical sensors to directly perform measurements.

## 6. Conclusions

This paper presents the development of a low-cost device for the monitoring of fatigue damage of structures subjected to vibrations in real-time. The device has been developed both in terms of hardware and software parts using low-cost commercial electronic elements and an ad-hoc developed algorithm.

To check the accuracy of the proposed device to supply correct results in real-time, a set of vibration tests were performed on a simple Y-shaped specimen excited with different random profiles until the failure occurs. The fatigue life of the component was monitored with the proposed device and the obtained results in terms of fatigue life were compared with those calculated in post-processing by monitoring the drop of the natural frequency of the specimen. The comparison shows that there is a good agreement between what is obtained with the proposed device and with the traditional method. The maximum error is always lower than 20%. The estimated fatigue life is always shorter than the actual and, being conservative, it can be considered acceptable. The validation of the device, however, has been conducted on a simple laboratory specimen. For this reason, the results presented in this study can be considered as a good starting point. Further developments and subsequent validation on more complex and realistic structures must be conducted in order to certify the accuracy of the proposed device/method and it will be the focus of future activities. However, the obtained outcomes indicate that the fatigue damage monitoring with low-cost instruments is viable, although further developments are still needed.

## Figures and Tables

**Figure 1 sensors-23-05143-f001:**
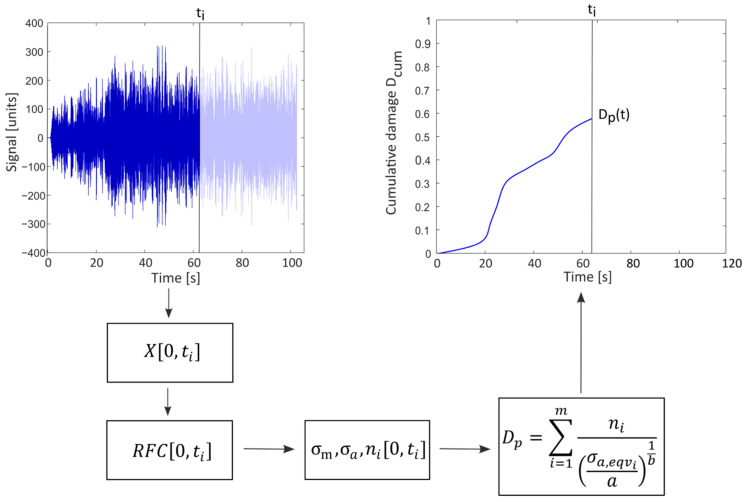
Flowchart of the classical method for the evaluation of fatigue damage in the time domain.

**Figure 2 sensors-23-05143-f002:**
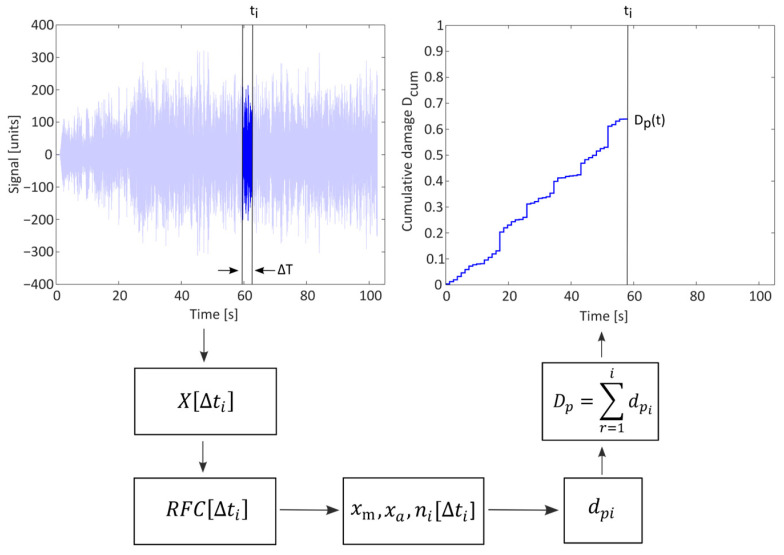
Flowchart of damage evaluation technique usable in real-time.

**Figure 3 sensors-23-05143-f003:**
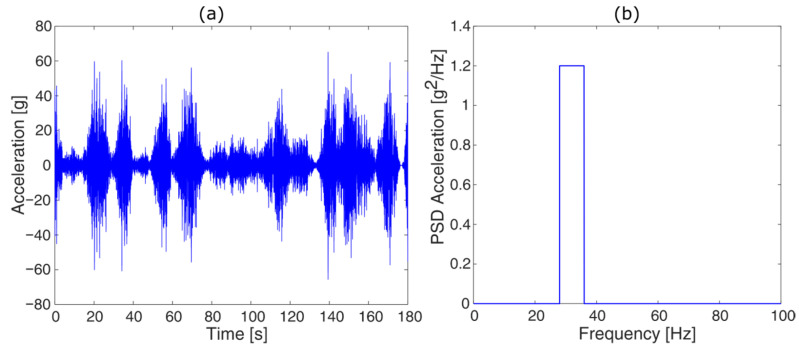
Non-stationary signal: (**a**) Time history; (**b**) associated PSD.

**Figure 4 sensors-23-05143-f004:**
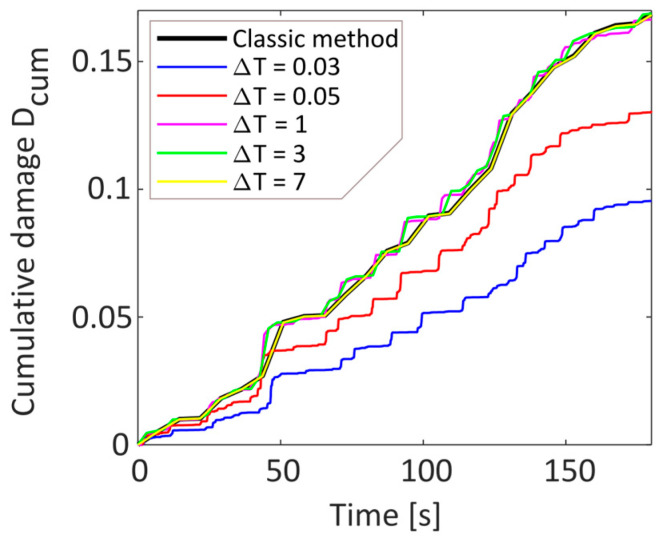
Comparison of the cumulative damage of different window durations with the classical method.

**Figure 5 sensors-23-05143-f005:**
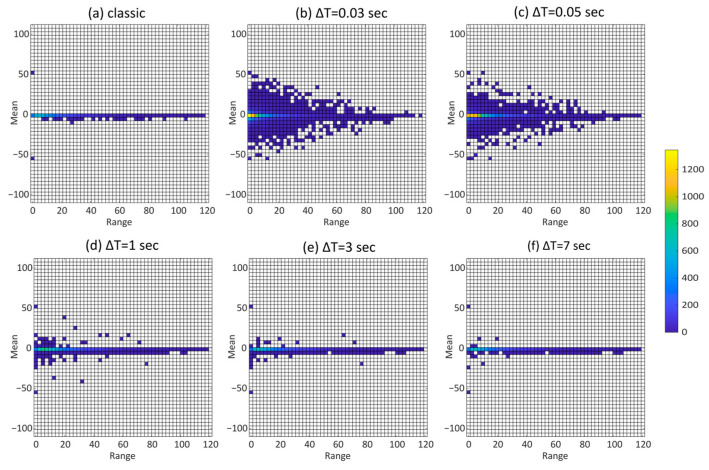
Comparison between Rainflow matrices obtained with the standard approach and the one proposed for different window durations.

**Figure 6 sensors-23-05143-f006:**
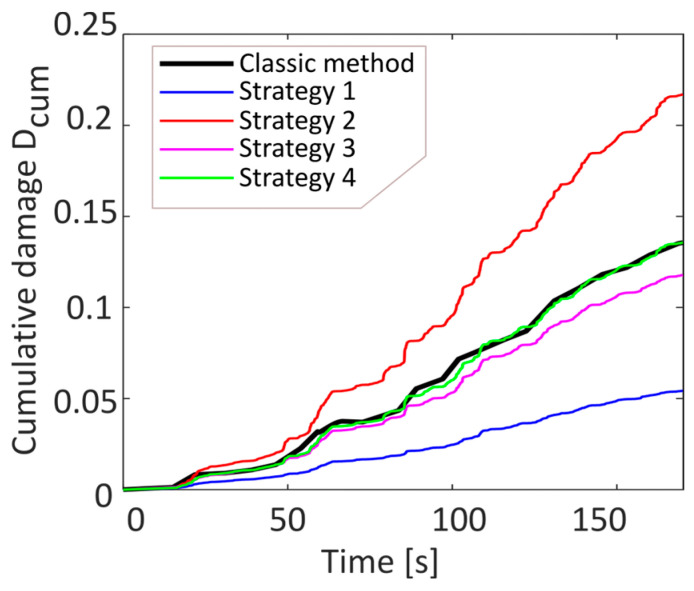
Comparison of the cumulative damage of different residue management strategies with the classical method for the non-stationary signal type.

**Figure 7 sensors-23-05143-f007:**
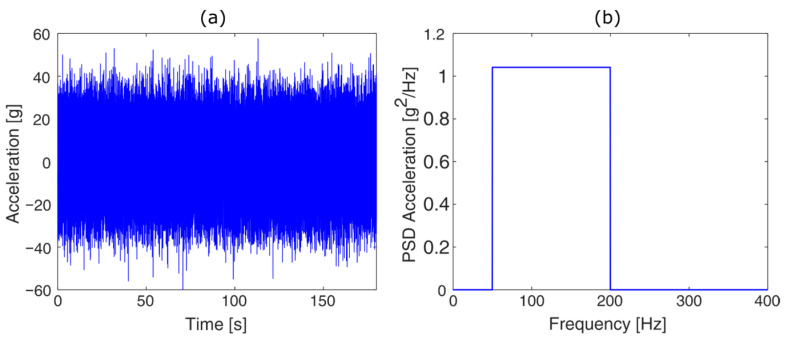
Wideband signal: (**a**) Time history; (**b**) associated PSD.

**Figure 8 sensors-23-05143-f008:**
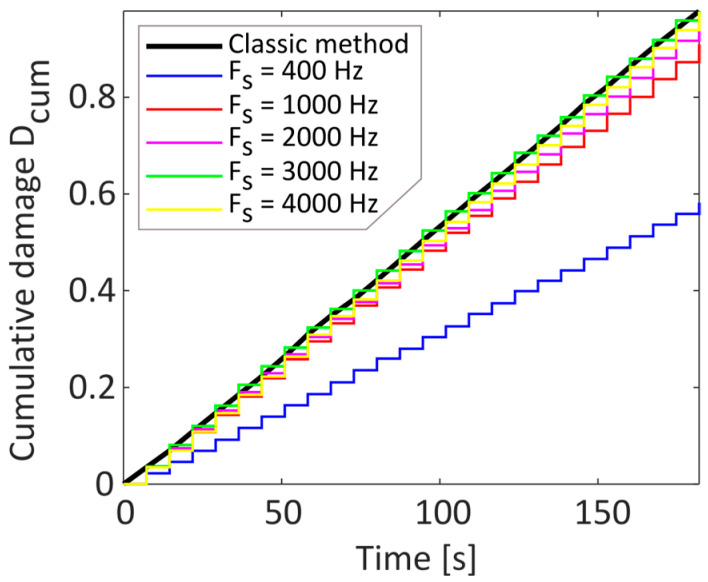
Comparison of the cumulative damage of sample frequency variation with the classical method for the wideband signal only.

**Figure 9 sensors-23-05143-f009:**
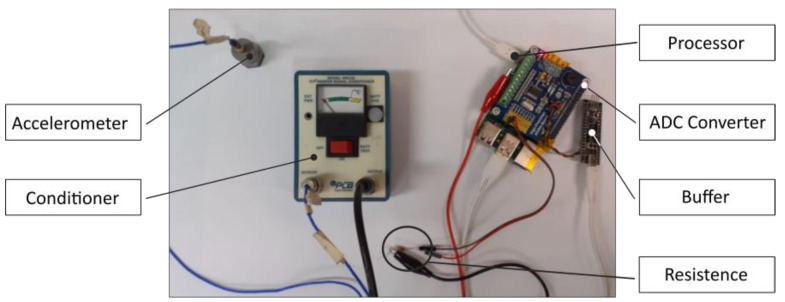
The prototype of the device realized in this activity to monitor the fatigue damage in real-time [34].

**Figure 10 sensors-23-05143-f010:**
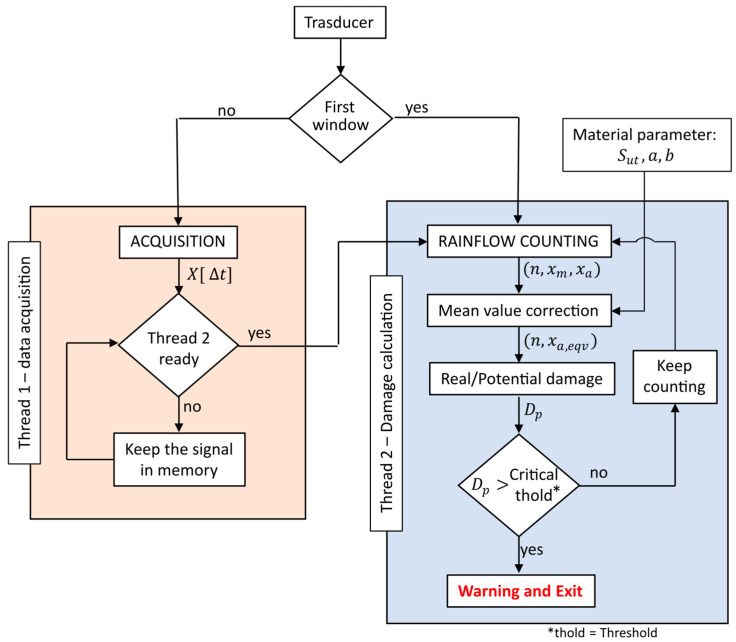
Multi-threading logic to calculate fatigue damage in real-time [34].

**Figure 11 sensors-23-05143-f011:**
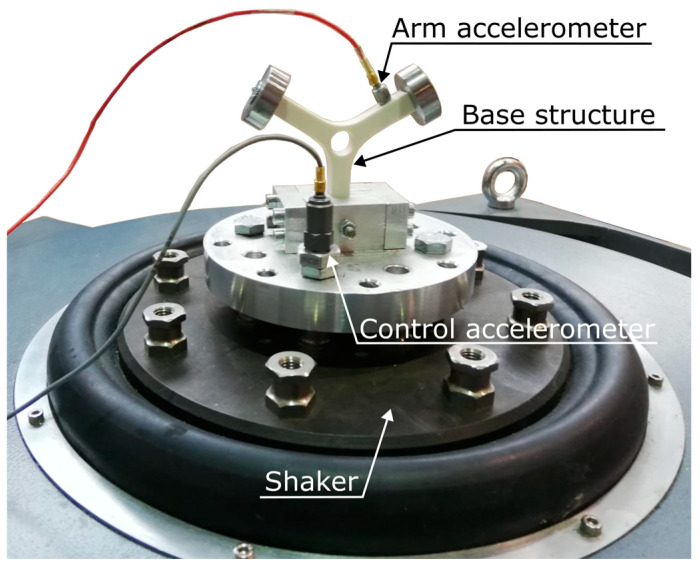
Experimental setup.

**Figure 12 sensors-23-05143-f012:**
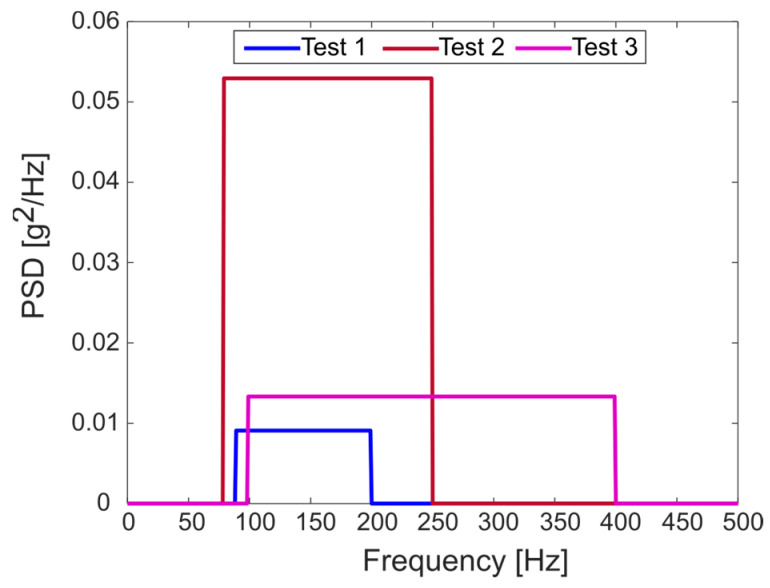
Waveform of the adopted PSD in the experimental tests.

**Figure 13 sensors-23-05143-f013:**
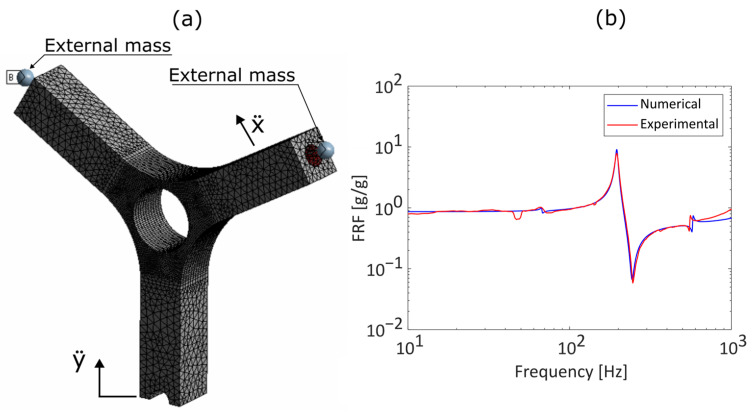
(**a**) FE model of the Y-shaped specimen; (**b**) experimental comparison of numerical and experimental acceleration frequency response function.

**Figure 14 sensors-23-05143-f014:**
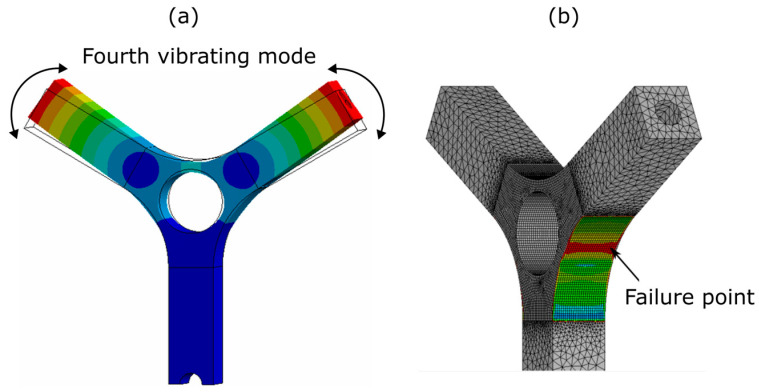
(**a**) Excited vibrating mode; (**b**) location of the failure point.

**Figure 15 sensors-23-05143-f015:**
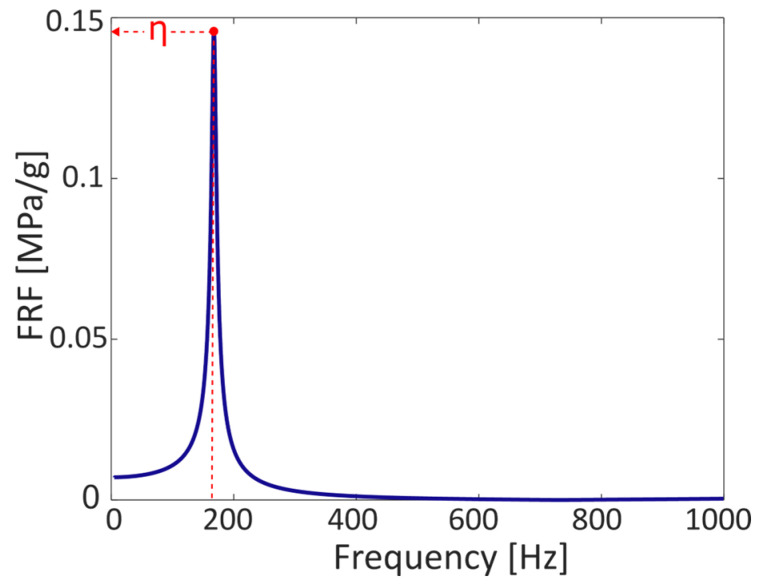
Frequency response function between the stress at the failure point and the acceleration measure on the arm of the specimen.

**Figure 16 sensors-23-05143-f016:**
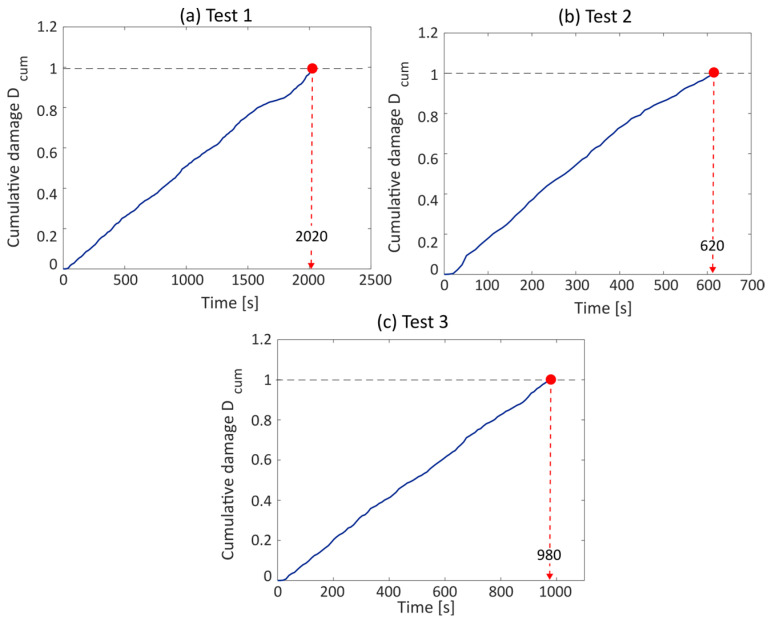
Estimated life evaluation for the three tests performed using the realized device.

**Figure 17 sensors-23-05143-f017:**
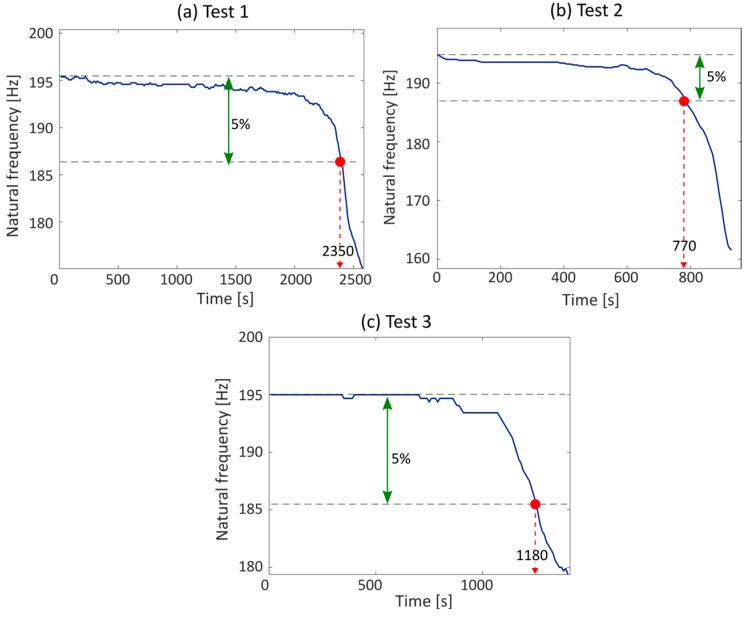
Fatigue life obtained by monitoring the frequency drop in post-processing for the three conducted experimental tests.

**Table 1 sensors-23-05143-t001:** Window duration variation analysis results for the non-stationary signal only.

Min. Frequency	Scale Factor	Window Duration	Total Damage	Percentage Error
[Hz]	[-]	[s]	[-]	[%]
28	1	0.03	0.0954	44
28	1.5	0.05	0.1306	23
28	28	1	0.1664	2
28	85	3	0.1687	1
28	190	7	0.1696	0.17

**Table 2 sensors-23-05143-t002:** Results for each considered signal from the window duration variation analysis.

Min. Freq.	Scale Factor	Win. Duration	Total Dam	Perc. Error
[Hz]	[-]	[s]	[-]	[%]
Non-stationary signal in section (Damage with standard method = 0.07859)				
20	1	0.05	0.0659	16.15
20	2	0.1	0.0709	9.81
20	30	1.5	0.0778	1.01
20	50	2.5	0.0781	0.56
20	100	5	0.0784	0.25
Non-zero mean signal (Damage with standard method = 0.0140)				
0.4	1	2.5	0.0133	4.71
0.4	2	5	0.0139	1.07
0.4	30	75	0.0142	1.14
0.4	50	125	0.0141	0.71
0.4	100	250	0.0141	0.71
Wideband signal (Damage with standard method = 0.9782)				
50	1	0.02	0.8265	15.51
50	2	0.04	0.8976	8.24
50	30	0.6	0.9922	1.43
50	50	1	0.9950	1.72
50	100	2	0.9769	0.13
Bimodal signal (Damage with standard method = 0.1608)				
10	1	0.1	0.1357	15.61
10	2	0.2	0.1452	9.70
10	30	3	0.1565	2.67
10	50	5	0.1556	3.23
10	100	10	0.1560	2.98

**Table 3 sensors-23-05143-t003:** Analysis of residue management strategy results for the non-stationary signal only.

	Residue Management Technique	Total Damage	Percentage Error [%]
1	Not counted	0.0608	64%
2	Counted as full cycle	0.2784	64%
3	Moved to next window	0.1560	8%
4	Counted as half a cycle	0.1696	0.17%

**Table 4 sensors-23-05143-t004:** Results for each considered signal from the residue management strategy analysis.

	Residue Management Technique	Total Damage	Percentage Error [%]
Non-stationary signal in section (Damage with standard method = 0.07859)			
1	Not counted	0.06829	13.11
2	Counted as full cycle	0.1072	36.40
3	Moved to next window	0.0795	1.16
4	Counted as half a cycle	0.0784	0.25
Non-zero mean signal (Damage with standard method = 0.0140)			
1	Not counted	0.00744	46.86
2	Counted as full cycle	0.0206	47.14
3	Moved to next window	0.0142	1.43
4	Counted as half a cycle	0.0141	0.71
Wideband signal (Damage with standard method = 0.9782)			
1	Not counted	0.916	6.39
2	Counted as full cycle	1.409	44.04
3	Moved to next window	1.029	5.19
4	Counted as half a cycle	0.977	0.13
Bimodal signal (Damage with standard method = 1608)			
1	Not counted	0.1323	17.71
2	Counted as full cycle	0.2128	32.33
3	Moved to next window	0.1697	5.55
4	Counted as half a cycle	0.1560	2.98

**Table 5 sensors-23-05143-t005:** Sample rate variation analysis results for the wideband signal only.

Max. Frequency	Scale Factor	Sampling Rate	Time Resolution	Total Damage	Error
[Hz]	[-]	[Hz]	[s]	[%]	[%]
200	2	400	$2.5\times 10^{-3}$	0.5820	41
200	5	1000	$1.0\times 10^{-3}$	0.9096	7
200	10	2000	$5.0\times 10^{-4}$	0.9558	2
200	15	3000	$3.3\times 10^{-4}$	0.9988	2
200	20	4000	$2.5\times 10^{-4}$	0.9769	0.13

**Table 6 sensors-23-05143-t006:** Results for each considered signal from the sample rate variation analysis.

Max. Frequency	Scale Factor	Sampling Rate	Time Resolution	Total Damage	Error
[Hz]	[-]	[Hz]	[s]	[%]	[%]
Non-stationary signal in section (Damage with standard method = 0.07859)					
100	2	200	5.00 × 10^−3^	0.0526	33.08
100	5	500	2.00 × 10^−3^	0.07301	7.11
100	10	1000	1.00 × 10^−3^	0.077311	1.64
100	15	1500	6.67 × 10^−4^	0.07807	0.67
100	20	2000	5.00 × 10^−4^	0.0784	0.25
Non-zero mean signal (Damage with standard method = 0.0140)					
20	2	40	2.50 × 10^−2^	0.0075	46.79
20	5	100	1.00 × 10^−2^	0.0104	25.86
20	10	200	5.00 × 10^−3^	0.0121	13.75
20	15	300	3.33 × 10^−3^	0.0141	0.71
20	20	400	2.50 × 10^−3^	0.0141	0.71
Non-stationary signal in section (Damage with standard method = 0.1699)					
36	2	72	1.39 × 10^−2^	0.0179	89.46
36	5	180	5.56 × 10^−3^	0.0482	71.63
36	10	360	2.78 × 10^−3^	0.1496	11.95
36	15	540	1.85 × 10^−3^	0.17655	3.91
36	20	720	1.39 × 10^−3^	0.1696	0.17
Bimodal signal (Damage with standard method = 1608)					
110	2	220	4.55 × 10^−3^	0.1064	33.83
110	5	550	1.82 × 10^−3^	0.14501	9.82
110	10	1100	9.09 × 10^−4^	0.1537	4.42
110	15	1650	6.06 × 10^−4^	0.15602	2.97
110	20	2200	4.55 × 10^−4^	0.156	2.98

**Table 7 sensors-23-05143-t007:** Comparison between the estimated fatigue life with the proposed device and the standard approach.

Test ID	Frequency Range	Input RMS	Estimated Life	Real Life	Percentage Error
[-]	[Hz]	[g]	[s]	[s]	[%]
1	90–200	1	2020	2350	16.3
2	80–250	3	620	770	19.04
3	100–400	2	980	1180	17.1

## Data Availability

The data that support the findings of this study are available on request from the corresponding author, [M.P.].

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
