# Peer review of "Development and Validation of a Low-Cost Device for Real-Time Detection of Fatigue Damage of Structures Subjected to Vibrations"

_sensors, 2023, doi:10.3390/s23115143_

Round 1
Reviewer 1 Report
Based on my 15 years experience in the academic and industrial research in the field of fatigue and fracture, I believe that the authors tried to provide a low-cost device and method to detect the fatigue damage in the structures subjected to the vibrations. So, I believe that this research is the main topic that are challenge in Industry to detect damage before suddenly fail and catastrophic accident. Also, the reported results can be used in different industries, such as energy. For example, one cause of failure in hydropower plant is mechanical damages induced vibrations. In addition, the article is well written and well structured. The written language is good and clear. The conclusions are consistent with the evidence and arguments presented. The references are appropriate. However, I believe that this manuscript should be edited before publishing. Therefore, please attention to the following points:
1- Page 8 and line 311, it seems that "in the first testing phase," is true and there is a typo.
2- Page 13 and lines 416 and 418, it seems that "figure 2" should change to "Figure 9".
3- Related to section 5 and page 15, finite element simulation performed in the present research should be describe in details, including materials, loading, analysis type, mesh convergence, etc. and it is better to place in the appendix.
4- Compared to previous techniques, this device is simple and low-cost. But, the used geometry is not in the standard and need more studies. In fact, this paper can be a primarily research and be a good starting points for future research.
5- In this specimen type, It seems to consider load as vibrations in shake, but in real, they should consider multiaxial fatigue loading by vibrations. So, a simple shacking device is not real. In industries, the components are subjected to more complex non-proportional loading.
Reviewer 2 Report
This paper presents a valuable contribution to structural health monitoring by developing a low-cost, real-time fatigue damage detection device for structures subjected to vibrations. The device, combining hardware and a signal processing algorithm, demonstrates accurate detection of structural damage through well-designed experiments on a Y-shaped specimen. The low-cost and ease of implementation make this device promising for various industrial sectors, with potential to reduce maintenance costs, enhance safety, and extend structural service life. The research exhibits a rigorous methodology and provides compelling evidence supporting the device's efficacy, paving the way for further exploration and wider adoption in the field.
In general, the paper is well-structured and seems to be contributing to the sensors Journal. However, the following minor revisions must be addressed by the authors before proceeding further:
(1) In the introduction section, the discussion about the detection techniques of fatigue damage should be expanded. Some recent development on structural health monitoring and damage detection may be included (e.g., DOI: 10.3390/s20113197)
(2) The title of section 2.1 should be deleted.
(3) It is not enough to provide the results of only one case in Section 3.1 of the paper. The results of other cases should be supplemented for comparison.
(4) The subheadings of section 4 should be changed to “4.1 The hardware” and “4.2 The software”.
Reviewer 3 Report
This paper developed a low-cost device for real-time detection of fatigue damage of structures subjected to vibrations. It provided a detailed introduction to the detection principle, hardware composition, and algorithm implementation of the device. The device has been verified through Y-shaped test pieces. In my view, the structure of this article is relatively complete and the research has specified engineering value. However, there are still some problems to be improved, specific comments are as follows:
Q1: In Section 2, “Goodman's correction” is used. This application is not reflected in the following Equations. “such as that reported in Equation 5, it is possible to obtain an indication of the residual life of the component also by measuring physical quantities different from stress.” Do different physical quantities also require mean correction?
Q2: In introduction, more recent progress on fatigue under virbation should be strengthened, like ultrasonic fatigue test in “Development of a photomicroscope method for in situ damage monitoring under ultrasonic fatigue test. Int J Struct Integr, 2022, 13(2): 237-250” and fatigue under impact loads in “Evaluation and prediction of material fatigue characteristics under impact loads: review and prospects. Int J Struct Integr, 2022, 13(2): 251-277”;
Q3: In Section 3.1, “…The duration of the signal depends on the frequency content of the contributions that participate in the analyzed signal, especially the one with the lowest frequency…” “…Usually, it is set as a multiple of the process's maximum frequency. Also, in this case, it is known that to capture the highest frequency variations, it is generally accepted to use a sampling frequency at least 10 times larger than the maximum frequency of the process…” Please provide more references regarding these descriptions.
Q4: In Section 3.1.1 and 3.1.3, may the frequency range affect the duration of the window? Comparison results of multiple signals in different frequency ranges should be provided.
Q5: In Section 3, it would be more convincing to provide a comparison of the differences and advantages between the classical program and the proposed method.
Q6: In Fig. 10, please further improve the software framework process and clarify the details of input/output and parameter setting.
Q7: In Section 5, please provide waveform diagrams and PSD for three types of signals, like Fig.3 and 7. “… The used PSDs have the following frequency ranges: 90-200 Hz, 80-250 Hz, and 100-400 Hz, while the used RMS values were 1g, 3g, and 2g, respectively.”
Q8: In Fig. 14, please distinguish the symbol and Y. Besides, Please check all the chapter number.
Q9: In Section 5, please provide more references about “Considering the maximum value of the frequency response function is however too conservative since the system is not excited only at the resonance frequency but with a wideband signal.”
Q10: The conclusion section should be shortened and focused on the main points of current work.
The english usage is acceptable for this work.
Round 2
Reviewer 3 Report
The work has been well refined, it can be accepted as it is.